# Genetic Polymorphisms of *5-HT* Receptors and Antipsychotic-Induced Metabolic Dysfunction in Patients with Schizophrenia

**DOI:** 10.3390/jpm11030181

**Published:** 2021-03-05

**Authors:** Diana Z. Paderina, Anastasiia S. Boiko, Ivan V. Pozhidaev, Anna V. Bocharova, Irina A. Mednova, Olga Yu. Fedorenko, Elena G. Kornetova, Anton J.M. Loonen, Arkadiy V. Semke, Nikolay A. Bokhan, Svetlana A. Ivanova

**Affiliations:** 1Mental Health Research Institute, Tomsk National Research Medical Center of the Russian Academy of Sciences, 634014 Tomsk, Russia; osmanovadiana@mail.ru (D.Z.P.); anastasya-iv@yandex.ru (A.S.B.); craig1408@yandex.ru (I.V.P.); irinka145@yandex.ru (I.A.M.); f_o_y@mail.ru (O.Y.F.); kornetova@sibmail.com (E.G.K.); asemke@tnimc.ru (A.V.S.); bna909@gmail.com (N.A.B.); ivanovaniipz@gmail.com (S.A.I.); 2Research Institute of Medical Genetics, Tomsk National Research Medical Center of the Russian Academy of Sciences, 634050 Tomsk, Russia; anna.bocharova@medgenetics.ru; 3Siberian State Medical University, 634050 Tomsk, Russia; 4Unit of PharmacoTherapy, -Epidemiology & -Economics, Groningen Research Institute of Pharmacy, University of Groningen, 9713AV Groningen, The Netherlands

**Keywords:** schizophrenia, antipsychotics, body mass index, weight gain, metabolic syndrome, serotonin, genes, pharmacogenetics

## Abstract

Background: Antipsychotic-induced metabolic syndrome (MetS) is a multifactorial disease with a genetic predisposition. Serotonin and its receptors are involved in antipsychotic-drug-induced metabolic disorders. The present study investigated the association of nine polymorphisms in the four 5-hydroxytryptamine receptor (*HTR*) genes *HTR1A*, *HTR2A*, *HTR3A*, and *HTR2C* and the gene encoding for the serotonin transporter *SLC6A4* with MetS in patients with schizophrenia. Methods: A set of nine single-nucleotide polymorphisms of genes of the serotonergic system was investigated in a population of 475 patients from several Siberian regions (Russia) with a clinical diagnosis of schizophrenia. Genotyping was performed and the results were analyzed using chi-square tests. Results: Polymorphic variant rs521018 (*HTR2C*) was associated with higher body mass index in patients receiving long-term antipsychotic therapy, but not with drug-induced metabolic syndrome. Rs1150226 (*HTR3A*) was also associated but did not meet Hardy–Weinberg equilibrium. Conclusions: Our results indicate that allelic variants of *HTR2C* genes may have consequences on metabolic parameters. MetS may have too complex a mechanistic background to be studied without dissecting the syndrome into its individual (causal) components.

## 1. Introduction

Antipsychotics are important therapeutic agents for patients with schizophrenia, but long-term use of these drugs increases the risk of developing type 2 diabetes mellitus, hyperlipidemia, and hypertension [1,2,3,4,5,6]. These drugs are known to increase the prevalence of metabolic syndrome (MetS), which is a clustering of well-known cardiovascular risk factors and is known to be augmented in a variety of psychiatric disorders [6]. Apart from increasing the likelihood of serious cardiovascular and malignant pathologies, significant weight gain can also affect compliance and cause a decrease in the quality of life of patients with schizophrenia, since, in addition to the stigma of schizophrenia, there is the stigma of obesity [7].

Several G protein-coupled receptors, mainly dopamine, serotonin, and noradrenaline receptors, are traditional molecular targets for antipsychotics [8]. The efficacy of classical antipsychotics was mainly associated with the antagonism of dopamine type 2 (D2) receptors. Atypical antipsychotics are far more complex D2 receptor antagonists and act beyond D2 antagonism, involving other receptor targets that regulate dopamine and other neurotransmitters. Therefore, atypical antipsychotics have fewer adverse effects like parkinsonism, hyperprolactinemia, apathy, etc., which are all linked to the strong blockade of D2 receptors [9]. A variety of mechanisms contribute to how treatment with antipsychotic drugs results in metabolic syndrome [3,10,11]. Roughly three components can be distinguished, related to behavioral (reward-seeking, neuropsychoimmunological), hypothalamic endocrine (appetite/satiety), and peripheral (adipocytes, immune system, abdominal organs) regulation [10]. Serotonergic (serotonin, 5-Hydroxytryptamine, *5-HT*) neurotransmission appears to be involved in all three of them as well as in the pathophysiology of schizophrenia [12]. Circuits regulating the intensity of reward-seeking and distress-avoiding behaviors are controlled by the habenuloid complex via ascending monoaminergic terminals originating in the upper brainstem [13,14,15]. Mesencephalic *5-HT* neurons projecting to the striatum, prefrontal cortex, and amygdaloid complex increase the intensity of distress-avoiding behavior directly and also indirectly by inhibiting dopaminergic and adrenergic neurons [16,17,18]. Serotonergic neurotransmission also has a key role concerning obesity through both cerebral and peripheral mechanisms [11,19,20]. Moreover, most atypical antipsychotic drugs have a considerable affinity to certain *5-HT* receptors [18,21,22].

Seven types of *5-HT* receptors have been identified, all but one (*5-HT3*) being G-protein coupled receptors (GPCRs) [18,23]. When also considering subtypes, at least 13 of these GPCRs can be distinguished [23]. *5-HT1A* is a post-synaptic receptor in limbic forebrain structures and a somatodendritic autoreceptor of *5-HT* neurons of the raphe nuclei. *5-HT1A* is also expressed by cholinergic neurons within both the brain and the gastrointestinal system. *5-HT1A* inhibits neuronal firing and neurotransmitter release via Gi/o-protein-coupled K^+^ channels (GIRK channels) [23]. Receptors of type *5-HT2* are widely distributed within the brain [24] but also have an important role in vascular contractility (*5-HT2A*, *5-HT2B*), colonic motor function (*5-HT2B*), and voiding (5*-HT2C*) [18,23,25,26,27]. *5-HT2* subtypes are coupled to Gq/11, which increases inositol 1,4,5-triphosphate levels and facilitates neuronal depolarization but, due to the activation of GABAergic interneurons, frequently inhibit their targets [18]. A characteristic of *5-HT2C* (and, to a lesser extent, *5-HT2A*) is constitutive activity, which enables clozapine to have inverse agonistic effects [28]. *5-HT3* subtypes are expressed by central and peripheral neurons where they induce rapid depolarization/repolarization by the opening of non-selective cation channels [23]. *5-HT3* within the lower brainstem is involved in vomiting and consists of *5-HT3A* subunits, while *5-HT3* subtypes on the peripheral (autonomic and somatosensory) neurons have profound effects on the cardiovascular system and regulate motility and secretion throughout the whole gastrointestinal system [23]. This peripheral *5-HT3* consists of a heteromeric combination of *5-HT3A* and *5-HT3B* subunits.

Several *HTR* genes are involved in the regulation of metabolic homeostasis, including *HTR1B*, *HTR1F*, *HTR2A/2C*, *HTR3*, and *HTR6* [11,19,20]. As MetS has a significant genetic component [29,30], genes encoding for these *5-HT* receptors can be considered good candidates for genetic association studies particularly aimed at studying the molecular mechanism responsible for drug-induced weight gain. Most often studied are single-nucleotide polymorphisms (SNPs) of the *HTR2A* and *HTR2C* genes, since the effect of atypical antipsychotics on hypothalamic *5-HT2A/2C*, in particular, is believed to contribute to drug-induced weight gain [7,10,11,31]. Moreover, polymorphic variants of the *HTR2A* gene are associated with higher body mass index (BMI) [32], greater waist circumference, and other components of metabolic syndrome [33]. Single-nucleotide polymorphisms of the *HTR2C* gene are also associated with obesity, weight gain, and BMI [34,35,36,37,38]. In a study by Yuan et al. [38], several haplotypes of the promoter region of *HTR2C* were identified and associated with obesity and diabetes. A pharmacogenetic study of other genes of the serotonergic system (*HTR3A, HTR3B*) did not reveal associations with weight gain induced by antipsychotics [39].

Studying the association of functional genetic variants with specific clinical phenomena can be applied to elucidate their mechanistic background [40]. We applied this investigational method several times in attempts to clarify a possible role of *5-HT* neurotransmission in the mechanism of two side effects of (atypical) antipsychotic drugs: dyskinesia [41,42,43,44] and hyperprolactinemia [45]. Therefore, studying the contribution of *5-HT* neurotransmission in the mechanisms of MetS can be considered a logical next step. However, due to the complex involvement of genuine metabolic as well as endocrine, immunological, and behavioral mechanisms [46,47,48,49,50], body weight is also considered in this study.

## 2. Materials and Methods

### 2.1. Patients

This study was conducted according to the protocol approved by the Bioethical Committee of the Mental Health Research Institute of the Tomsk National Research Medical Center of the Russian Academy of Sciences (Protocol 187, approval on 24.04.2018). After obtaining informed consent we recruited 475 patients with schizophrenia being treated at the clinics of the Research Institute of Mental Health of the Tomsk National Research Medical Center, the Tomsk Clinical Psychiatric Hospital, the Hospital of the Siberian State Medical University, the Kemerovo Regional Clinical Psychiatric Hospital, and the N.N. Solodnikova Clinical Psychiatric Hospital of Omsk in the Russian Federation.

The main criteria for the inclusion of patients in the study were a verified diagnosis of schizophrenia according to ICD-10 (International Classification of Diseases, 10th revision) criteria as assessed by applying a structured clinical interview (Structured Clinical Interview for the DSM [SCID]), age 18–65 years, the patient’s informed consent, Caucasian appearance, and the absence of severe organic pathology or somatic disorders in the stage of decompensation.

The antipsychotic and concomitant therapy received at the time of the examination (drugs, dosages used, duration of current drug use) were assessed, as well as previous antipsychotic and concomitant somatic therapy during the preceding six months. We used the chlorpromazine equivalent (CPZeq) daily dosage to standardize the dose, efficacy, and side effects of antipsychotics [51].

MetS was diagnosed according to the criteria of the International Diabetes Federation (IDF, 2005) [52], including the definition of abdominal obesity (waist circumference more than 94 cm in men, or more than 80 cm in women) and the presence of any two of the following four signs:Concentration of triglycerides (TG) above 1.7 mmol/L, or lipid-lowering therapy;Concentration of high-density lipoproteins (HDL) of less than 1.03 mmol/L in men, or less than 1.29 mmol/L in women;Blood pressure (BP) greater than or equal to 130/85 mm Hg, or the usage of antihypertensive therapy;Concentration of glucose in blood serum greater than or equal to 5.6 mmol/L, or previously diagnosed type 2 diabetes mellitus.

### 2.2. Genetic Analysis

Blood samples for biochemical and pharmacogenetic studies were taken by antecubital venipuncture in vacutainer tubes with SiO_2_ as a clot activator (to obtain serum) or with EDTA (to isolate genomic DNA by the standard phenol–chloroform method).

Genotyping of nine single-nucleotide polymorphisms (SNPs) of genes of the serotonergic system *HTR1A* (rs1423691, rs878567), *HTR2A* (rs6314), *HTR3A* (rs2276302, rs1150226), *HTR2C* (rs1414334, rs521018, rs498177), and *SLC6A4* (rs16965628) was carried out using a MassARRAY^®^ Analyzer 4 mass spectrometer (Agena Bioscience ™) and a QuantStudio ™ 3D Digital PCR System Life Technologies amplifier (Applied Biosystems) using TaqMan Validated SNP Genotyping Assay kits (Applied Biosystems) based at The Core Facility “Medical Genomics”, Tomsk National Research Medical Center of the Russian Academy of Sciences.

### 2.3. Statistical Analysis

Statistical analysis was carried out using SPSS software, release 23.0. The Hardy–Weinberg equilibrium (HWE) of genotypic frequencies was tested by the chi-square test. Pearson’s chi-squared test was used for between-group comparisons of genotypic and allelic frequencies at a significance level of *p* < 0.05. Assessment of the association of genotypes and alleles of the studied polymorphic variants of genes with a pathological phenotype was carried out using the odds ratio (OR) with a 95% confidence interval for the odds ratio (95% CI).

## 3. Results

A total of 475 patients receiving long-term antipsychotic therapy were examined. Metabolic syndrome was diagnosed in 126 patients (26.5%). Table 1 presents the main demographic and clinical parameters of the studied patient groups.

In our sample, MetS was more often diagnosed in women with schizophrenia. The patients with MetS were significantly older (*p* < 0.0001), and the duration of illness in these patients was significantly longer than that in patients without MetS (*p* = 0.001). The study groups also showed significant differences in body mass index (*p* < 0.0001).

Deviation from the HWE was found for the rs1150226 polymorphic variant of the *HTR3A* gene; hence, this polymorphism was excluded from further consideration. As the *HTR2C* gene is located on the X chromosome, its polymorphic variants should not meet HWE. Taking this localization into account, the distributions of the genotype and allele frequencies of the studied *HTR2C* genes (rs1414334, rs521018, and rs498177) were analyzed separately in the groups of women (*n* = 214) and men (*n* = 261).

There were no statistically significant differences in any of the eight studied SNPs of genes of the serotonergic system and metabolic syndrome in patients with schizophrenia receiving antipsychotic therapy for a long time. Statistically significant associations were, however, revealed in groups of patients with BMI of <25 and BMI of >25 (Table 2).

Statistically significant differences in the distribution of genotype frequencies were found for the polymorphic variant rs521018 of the *HTR2C* gene in the group of women with schizophrenia (*p* = 0.033). Carriage of the heterozygous genotype GT causes a predisposition to increased weight gain in women receiving antipsychotic treatment (OR 1.97; 95% CI: 1.09–3.55).

## 4. Discussion

The complexity of *5-HT* signaling lies in the large number of receptor genes encoding for seven main *5-HT* receptor types with several further subtypes, dimerization with other receptor proteins, and RNA editing and alternative splicing of receptor transcripts [20,23,53,54]. Several serotonin receptors are involved in the regulation of metabolic homeostasis, such as *5-HT1A*, *5-HT2A*, *5-HT2C*, *5-HT3*, and *5-HT6* [7,11,20]. Agonists of *5-HT1A* and *5-HT2C* have opposite effects on food intake: *5-HT1A* receptors increase food intake, while *5-HT2C* receptors decrease appetite [55,56]. The selective *5-HT2C* agonist lorcaserin has been approved in the USA for supportive treatment in weight management [57]. Among all *5-HT* receptors, the *5-HT2C* are most strongly involved in the pharmacological action of serotonin [54]. Particularly, *5-HT2C* expressed by a subset of pro-opiomelanocortin neurons in the hypothalamic arcuate nucleus and the brainstem nucleus of the solitary tract have a crucial role in mediating anti-orexigenic activity [56,58]. In humans, antagonization of the *5-HT2C* receptor by atypical antipsychotics such as clozapine and olanzapine leads to weight gain [59,60,61].

In our study, we failed to establish an association between eight *HTR* genotypes and MetS. This may be due to the limited number of patients with MetS, but it is more likely due to the complex involvement of serotonergic transmission as well as other neurotransmitters in the mechanisms of the separate components of MetS. When studying one of these (indirect) components, we could find a relationship between rs521018 of *HTR2C* and increased body weight. 

Many allelic variants of the *HTR2C* gene have been studied in the context of weight gain induced by antipsychotic therapy. These include rs6318*G, rs3813928*A, rs1414334*C, rs498207*A, rs518147*C, rs498177*C, and rs521018*T. A study by Mulder and colleagues showed that the rs1414334*C allele is associated with MetS in patients taking clozapine (OR 9.20; 95% CI 1.95–43.45) or risperidone (OR 5.35; 95% CI 1.26–22.83) [62]. In a study by Bai et al., the polymorphic variant rs498177 showed a significant association with MetS in female patients, and allele C was associated with an increased risk of MetS (*p* = 0.0007) [63]. However, the haplotype of the polymorphic variants rs521018*A and rs498177*C in the *HTR2C* gene significantly reduces the risk of MetS (adjusted *p* = 0.0108) in women [63].

Unfortunately, rs1150226 of *HTR3A* did not meet the HWE criterion as the A allele was significantly associated with higher BMI (data not shown). Antipsychotic drugs can also interfere with the expression of various receptors and the release of neurotransmitters. Polymorphic variants of the *HTR3A* gene (rs2276302, rs1062613, and rs1150226) have been shown to be associated with the response to clozapine therapy [64], which may contribute to weight gain [37,60]. Antipsychotics such as clozapine and olanzapine cause significant metabolic overload but are considered effective treatments for patients who do not respond to other therapies. Therefore, rs1150226 should be studied again in an independent patient population.

Further study of the molecular genetic factors of MetS and the mechanisms by which antipsychotics affect metabolic parameters is necessary to assess the risk of metabolic disorders and the implementation and individual approach to therapeutic tactics. It is possibly useful to expand the studied sample and to include in the analysis other risk factors for the development of MetS, such as the antipsychotic therapy used, smoking, and lifestyle, as well as to study individual components of MetS in patients.

## 5. Conclusions

Our study did not show an association of serotonin receptor genes with MetS in patients with schizophrenia. However, associations of one of the studied polymorphic variants with an increased BMI were revealed. Metabolic syndrome is a complicated symptoms complex that consists of separate components, including obesity, dyslipidemia, impaired glucose metabolism, and arterial hypertension. Serotonin is involved in these pathophysiological processes to varying degrees. We suggest that the involvement of serotonergic neurotransmission in MetS should be better studied after dissecting the syndrome into its individual (causal) components.

## Figures and Tables

**Table 1 jpm-11-00181-t001:** Demographic and clinical parameters of the studied patient groups.

Parameter		Patients without MetS, *n* = 349 (73.5%)	Patients with MetS, *n* = 126 (26.5%)	*p*-Value
Gender	Women	142 (40.7%)	72 (57.1%)	0.001
	Men	207 (59.3%)	54 (42.9%)	
Age, years M ± SD		38.72 ± 11.43	43.75 ± 11.72	<0.0001
duration of illness, years Me [Q1; Q3].		12.0 [6.0; 20.0]	17.0 [9.0; 22.0]	0.001
CPZeq, dose Me [Q1; Q3].		400.0 [225.0; 750.0]	400.0 [203.0; 741.0]	0.919
Body mass index (BMI) M ± SD		24.45 ± 4.83	30.45 ± 6.36	<0.0001

Note. Me [Q1; Q3]—median and quartiles (first and third); MetS: metabolic syndrome; CPZeq: chlorpromazine equivalent; M ± SD—mean and standard deviation.

**Table 2 jpm-11-00181-t002:** Distributions of alleles and genotypes of *HTR2C* single-nucleotide polymorphism (SNPs) in groups of patients with body mass index (BMI) values of <25 and >25.

SNP	Genotypes, Alleles	BMI < 25	BMI > 25	OR	95% CI	χ^2^	*p* Value
*HTR2C* rs521018	GG	12 (13.8)	5 (5.2)	0.34	0.12–1.02	6.85	0.033
	GT	38 (43.7)	58 (60.4)	1.97	1.09–3.55		
	TT	37 (42.5)	33 (34.4)	0.71	0.39–1.29		
	G	62 (35.6)	68 (35.4)	0.99	0.65–1.52	0.00	0.965
	T	112 (64.4)	124 (64.6)	1.01	0.66–1.55		

## Data Availability

The datasets generated for this study will not be made publicly available, but they are available on reasonable request to Svetlana A. Ivanova (ivanovaniipz@gmail.com), following approval of the Board of Directors of the MHRI, in line with local guidelines and regulations.

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
