# Peer review of "Genetic Polymorphisms of 5-HT Receptors and Antipsychotic-Induced Metabolic Dysfunction in Patients with Schizophrenia"

_jpm, 2021, doi:10.3390/jpm11030181_

Round 1

Reviewer 1 Report

I could read your paper and have some small suggestion and correction to be made before publication.

At line 61 I would add a couple of sentences about the role of the many target receptors of atypical antipsychotics in their efficacy and adverse reactions (include good reviews such as Aringhieri et al., 2018; Stępnicki et al., 2018) before writing about the 5-HT system specifically.

Be careful with the names of receptors and genes. At line 74 you should replace HTR which refers to genes with 5-HT receptors, which is the name of proteins. The same is for HTR3 which chould be 5-HT3, and all through the paper when you refer to proteins. At line 75 g-protein should be in capital “G-protein”.

Line 81: “important role in, e.g., vascular…” not clear in English, please remove the e.g. and make it “important role in vascular…”

Line 85: “often finally” is not clear in English, reformulate

Line 86: again remove e.g.

In table 1 correct the last line BMI where you have M+SD and change with M±SD. Line 179: in the notes of table 1 insert also the M±SD as abbreviation, although more common than median and quartiles, to have uniformity.

Line 187: “it not meeting HWE for three poly-morphic variants can be ignored” not clear in English please reformulate

Line 214: correct with “the most strongly”

Line 222: include in the sentence that the lack of association is due to the complex involvement of serotonin system as well as other neurotransmitters in MetS.

Line 224: the tone of “evident” is too much enthusiastic; you could find an association only in a subgroup of your population, I would re-write “a relationship between rs521018 of HTR2C and increased body weight became evident” with something like “we could find/describe a relationship between rs521018 of HTR2C and increased body weight”.

Line 239: remove the citation “Souza et al., 2010”

Reviewer 2 Report

The article by Paderina et al. investigated the putative relationship between several 5-HT receptors and metabolic syndrome (MetS) in the context of schizophrenia and antipsychotic-induced MetS. The topic is relevant, and, although the authors report a lack of association between MetS and these 5-HT polymorphisms, the manuscript adds knowledge to an intricate field and should be taken into account in the future. The overall style and language are adequate. There are, however some minor concerns that the authors might want to consider so as to increase the impact of their research.

·Introduction:
   -In page 2, line 71, authors state "a key role in regulating obesity". They might consider using a more straighforward term, such as "regulating weight" or "regulating body composition", or simply leave the sentence as "a key role concerning obesity".
   -Although the choice of the literature used for justifying the relevance of the 5-HT receptor subtypes is adequate, the manuscript would highly benefit from a more detailed characterization of the relationship between 5-HT and schizophrenia, even in absence of metabolic syndrome (for instance: Selvaraj et al. 2014: Alterations in the serotonin system in schizophrenia: A systematic
review and meta-analysis of postmortem and molecular imaging
studies).

·Materials and methods + Results:
   -It would be useful that the authors reported, when appropriate, effect size, along with the explanation of the index used and the small/medium/large sizes according to Cohen (1988).

·Discussion:
   -As the authors state, future studies investigating the relationship between 5-HT receptors and the individual components of MetS are crucial for better understanding the phenomenon. Although it is still a question to be addressed, however, a brief, speculative paragraph going in-depth into this idea might be positive for a more complete discussion.
